# Protocol for development and validation of a context-appropriate tool for assessing organisational readiness for change in primary health clinics in South Africa

Carrie Brooke-Sumner,[1] Katherine Sorsdahl,[2] Carl Lombard,[3,4]
Petal Petersen-Williams,[1,5] Bronwyn Myers[1,5]

[1]Alcohol, Tobacco and Other Drug Research Unit, South African Medical Research Council, Cape Town, South Africa
[2]Alan J Flisher Centre for Public Mental Health, University of Cape Town, Cape Town, South Africa
[3]Biostatistics Unit, South African Medical Research Council, Cape Town, South Africa
[4]School of Public Health and Family Medicine, University of Cape Town, Cape Town, South Africa
[5]Department of Psychiatry and Mental Health, University of Cape Town, Cape Town, South Africa

**Correspondence to**
Dr Carrie Brooke-Sumner;
carrie.brooke-sumner@mrc.ac.za

## ABSTRACT

**Introduction** A large treatment gap for common mental disorders (such as depression) exists in South Africa. Comorbidity with other chronic diseases, including HIV and diseases of lifestyle, is an increasing public health concern globally. Currently, primary health facilities as points of care for those with chronic disease provide limited services for common mental disorders. Assessing organisational readiness for change (ORC) towards adopting health innovations (such as mental health services) using contextually appropriate measures is needed to facilitate implementation of these services. This study aims to investigate the validity of the Texas Christian University Organisational Readiness for Change (TCU-ORC) scale in the South African context. Subsequently, we will develop a shortened version of this scale. This study is nested within Project MIND, a multiyear randomised controlled trial that is testing two different approaches for integrating counselling for common mental disorders into chronic disease care. Although the modified, contextually appropriate ORC measure resulting from the proposed study will be developed in the context of integrating mental health into primary healthcare services, the potential for the tool to be generalised to further understanding barriers to any change being implemented in primary care settings is high.

**Methods and analysis** We will establish internal consistency (Cronbach's alpha coefficients), test-retest reliability (intraclass correlation coefficient) and construct validity of the long-form TCU-ORC questionnaire. Survey data will be collected from 288 clinical, management and operational staff from 24 primary health facilities where the Project MIND trial is implemented. A modified Delphi approach will assess the content validity of the TCU-ORC items and identify areas for potential adaptation and item reduction.

**Ethics and dissemination** Ethical approval has been granted by the South African Medical Research Council (Protocol ID EC004-2-2015, amendment of 20 August 2017). Results will be submitted to peer-reviewed journals relevant to implementation and health systems strengthening.

### Strengths and limitations of the study

► The study leverages and contributes to an existing partnership between the research group and the Department of Health which enables collection of survey data from participants (health facility staff) who would otherwise be difficult to access.

► The study methods enable a tangible output for health facility staff in the form of the short-form tool that can enable assessment of barriers to change in their contexts.

► Collection of survey data from only the Western Cape province may limit generalisability, given that it is considered better resourced than most other provinces. However, the planned Delphi process will involve experts from various provinces in the country.

► Reaching the required sample size may be challenging given the time constraints on health facility staff in South African primary health settings. A proactive engagement and follow-up plan with facility staff will be used to maximise the number of completed surveys collected.

## BACKGROUND

South Africa, like high-income countries (HIC) and increasingly other low/middle-income countries (LMIC), faces a burgeoning epidemic of non-communicable diseases including mental, neurological and substance use (MNS) disorders. Addressing this is a necessary priority for effective service delivery within the health sector.[1] Not only do MNS disorders rank third in contribution to overall disability-adjusted life years in South Africa,[2] but these disorders are associated with acquisition of communicable (eg, HIV) and non-communicable diseases (eg, diabetes)[3 4] and contribute to morbidity and mortality associated with these conditions. Despite the burden of disease associated with untreated MNS disorders, the number

of people with untreated MNS disorders remains large in South Africa—less than a quarter of people living with these conditions ever receive any form of treatment.[5 6] This treatment gap is larger for MNS disorders such as depression and substance use disorders than for severe disorders (such as schizophrenia). For example, the median treatment gap across countries for schizophrenia has been estimated at 32.2% compared with 56.3% for depression.[7 8] Given this variation, prioritising specific conditions that display the largest unmet need is warranted.

South Africa has made important progress towards the provision of accessible mental healthcare through the development of the comprehensive Mental Health Policy and Strategic Framework (2013–2020). In this policy, a task-sharing approach for delivery of mental health services through the existing chronic disease care platform in primary healthcare (PHC) is proposed as a potentially feasible way of reducing the mental health treatment gap for people with chronic diseases.[9 10] In this approach, responsibility for screening for MNS disorders (eg, depression and alcohol use disorder, which can be common in those with other chronic diseases), delivery of brief counselling for these disorders and, if needed, referral to more specialist mental healthcare is devolved from mental health providers to non-specialist cadres of health workers such as non-specialist nurses and facility-based lay counsellors.[10]

There has been limited progress in implementing this integrated care approach, perhaps due to a lack of resources being directed towards this additional mental health service.[11] This challenge is not limited to the mental health arena, since PHC facilities face many staff and other resource shortages that make implementation of any new service challenging. In addition, although the policy imperative for the roll-out of integrated mental health services exists,[12] this integrated approach requires a new and 'additional' aspect to work for front-line workers in clinics, who may find this threatening or stressful.[13] Many individuals are resistant to change and the health system itself as an organisation valuing stability is resistant to change.[14] Consequently, the degree to which organisations are ready for change to adopt new health services can be important predictors of whether these innovations are successfully adopted.[15 16] It is within this context that the study of organisational readiness for change (ORC) is developing in health systems research.

ORC is a complex and multifaceted concept, for which several theoretical models and frameworks have been put forward. These identify key components of ORC including organisational dynamics, climate and culture, change processes and individual organisational member characteristics.[17–19] ORC as defined by Weiner is a shared psychological state that relies on organisation members' motivation to change (change commitment) and belief in their own capacity to change (change efficacy).[18] While ORC is a shared psychological state, it is dependent on individual organisation members being willing and prepared for change, which is related both to characteristics of the individual as well as to the organisational components of ORC (eg, whether there is an organisational culture supportive of change in general). ORC is also seen as varying according to the value placed on the change by organisation members, as well as their perceptions of the environment in which the change will take place (including existing tasks, availability of resources and contextual factors).[18]

There is particular interest in conceptualising and measuring ORC in the health sector as organisations in which a higher degree of ORC is present may support individual members in cooperating and persevering towards implementing a change,[18] such as the adoption of a new service into routine practice. This in turn may lead to better service provision and subsequent health benefits to patients. With this conceptualisation of ORC, understanding barriers to ORC through organisations' self-assessment,[14] and intervening to address barriers,[20] is key to the sustainable adoption of new practices. The challenge is to find the best way of measuring ORC so that these barriers to change can be identified and addressed prior to implementing a new health service.[18 21 22]

Despite the drive to integrate MNS services into chronic disease care in LMICs, there has been little investigation of ORC in these settings. Questions about how to assess ORC accurately have hampered the study of organisational readiness for MNS services in LMIC health systems. There are a variety of well-developed scales for assessing ORC and other implementation factors in HIC contexts.[23] The Texas Christian University ORC (TCU-ORC) scale[24] is among the most widely used of the ORC scales, and has been tailored for use in health service research.[20] A recent systematic review covering ORC measures specifically highlighted that only 7 of the 43 measures reviewed had evidence for their validity and reliability.[25] It identified the TCU-ORC scale as having the strongest evidence for validity of all measures assessed. Studies in HIC contexts have shown that this measure has adequate internal consistency and good predictive validity for indicators of programme functioning, supporting the appropriateness of this measure for identifying functional barriers to organisational change.[24–26] This may be particularly relevant in the South African context since barriers to adoption of a new MNS service may not be specific barriers (eg, relating to the service itself), but rather general barriers related to the overall functioning of the facility. However, evidence in support of the TCU-ORC scale is HIC-specific and it is unclear whether the ORC items are relevant for LMIC contexts where resource allocation, leadership and other organisational dynamics within the public healthcare system are likely to be quite different from those in HIC settings. Although the TCU-ORC has been used in South Africa, with one study providing norms for this measure in the South African context,[27] little is known about the measure's psychometric properties in LMIC settings and South Africa specifically.[27] Consequently, there is a need for contextually validating this measure.

Further, the TCU-ORC in its current form is lengthy and time consuming to complete, comprising 125 items. Among measures used in implementation research, more than 60 items can be considered lengthy[23] and may lead to reduced response rates. While it is important to ensure that measures provide a depth of information across a broad range of ORC constructs, this should be balanced with the administrative burden longer measures may present to health managers. Measures that are brief and represent low administrative burden may be more feasible and acceptable for management staff in low-resource settings who are often drawn into clinical work due to high patient burden.[28] Developing a shortened version of the ORC may therefore enhance feasibility of assessing ORC within the context of South African PHC facilities.[29] The process described in this study therefore aims to enable a reduction of items in the measure, while retaining the ability to collect the essential information required to inform effective implementation of a new service, thus aiming to balance quality and brevity.

## METHODS
### Study aim
The overall goal of this study is to contribute to the documented need for contextually adapted and validated measures for ORC assessment (ORCA)[24] as an area of development for implementation science in a middle-income country context. More specifically, this study has two aims. First, the study aims to establish the internal consistency, test-retest reliability and construct validity of the long-form TCU-ORC questionnaire. Second, the study aims to develop a shortened version of the TCU-ORC that is feasible to use in South African PHC facilities. This can form a basis for assessing barriers and facilitators of adoption of new health interventions and guide the development of interventions to enhance the likelihood of effective implementation.

This study is nested within Project MIND, a multiyear study aiming to test two different approaches to service organisation for integrating counselling for depression and/or hazardous or harmful alcohol use into chronic disease care. Project MIND focuses on the provision of a facility-based lay counsellor-delivered programme for patients with depression and/or hazardous or harmful alcohol use who are receiving ongoing treatment for HIV or diabetes in the Western Cape province. Project MIND sites are 24 purposefully selected PHC facilities, with urban and rural representation.

### Measures
This study will collect data from health managers and providers using three measures of ORC. All of these measures are self-report measures designed for use by healthcare staff and managers. The TCU-ORC scale is a comprehensive measure of ORC as it measures individual psychological factors and structural (health system) factors[30] theoretically associated with ORC. It has four

domains: Motivational Readiness for Change (33 items), Institutional Resources (31 items), Staff Attributes (31 items) and Organisational Climate (30 items).[24] Each item is scored on a five-point Likert scale ranging from strongly disagree to strongly agree.

To assess convergent validity, two other measures with similar items will be used—the ORCA[31] and the Checklist for Assessing Readiness for Implementation (CARI).[32] The ORCA comprises three scales assessing (1) strength of evidence for the change to be introduced, (2) quality of the organisational context and (3) capacity for organisational facilitation of the change.[31] There is some evidence for the overall reliability and factor structure of the ORCA; however, some of the subscales in the evidence scale have inadequate reliability and two of the ORCA subscales did not load significantly onto any of the three factor scales above.[31] The CARI checklist includes assessment in eight key areas: organisational capacity, organisational climate or culture, staff capacity, system level capacity, functional considerations, senior leadership, training and implementation plan.[33] This second instrument was selected because it contains items that are broadly similar in content and wording to the TCU-ORC items although data on its psychometric properties remain to be published. In addition, relevant demographic and work-related data, including age, gender, education level, profession and time in profession, will be collected from participants to examine differential response to items.

### Participants and procedures
#### Aim 1: assessing internal consistency, construct validity and test-retest reliability of the TCU-ORC
Participants will be clinical staff from PHC facilities relevant to the implementation of Project MIND, as well as staff responsible for facility management and operations. These would include facility managers, operational managers, family physicians, medical officers, chronic care nurses, mental health nurses (where present), subdistrict managers (eg, PHC Managers), facility-based counsellors (employed by local non-governmental organisations) and their supervisors. We aim to secure participation of 12 appropriately selected staff members from each facility (288 participants from 24 facilities). Participants will be purposely selected to complete the assessment, using the principle of maximum variation sampling to promote representation of the required health personnel cadres and management staff. If it is possible to recruit sufficient numbers for each staff category, we will explore areas of consensus or disagreement on the ORC measure by analysing the data by role. Preliminary meetings will be held with facility managers from each facility to identify the relevant personnel. At a 'readiness workshop' that will introduce facility staff to the Project MIND trial, written informed consent to participate in this organisational assessment process will be obtained. Consenting individuals will then be asked to provide some contact information to facilitate follow-up before self-completing the demographic questionnaire

and three ORC questionnaires. They will be asked to complete the questionnaires after the workshop. One to 2 weeks after the workshop, participants will be asked to recomplete the questionnaires to enable test-retest reliability to be assessed. Participants will be offered a gift voucher to the value of R100 (~USD 10) to thank them for their time and participation.

## Aim 2: Delphi process for questionnaire reduction

We will use a modified Delphi approach drawing on a panel of South African experts relevant to health system strengthening and organisational change to assess the content validity of the TCU-ORC items and identify areas for potential adaptation, cultural modification and item reduction. The Delphi method is well suited to measure adaptation[34] and over the past decade has been used widely to gain expert consensus in health research.[35–37]

We will use snowball sampling to identify up to 30 experts to participate in the Delphi process. Experts will be identified within three panels—Panel 1 Academics and researchers; Panel 2 Governmental service providers and Panel 3 Non-governmental service providers. We will approach academics from a variety of institutions and across different provinces of South Africa. Service providers approached will be from the national level, as well as provincial and district level within the Western Cape province. The expert panels will therefore be broadly representative of the various regions in the country and will consist of individuals with experience in health systems research and development, as well as health service delivery.

The Delphi approach will involve electronic surveys to assess experts' opinions on the suitability of TCU-ORC items, providing anonymised reports on the aggregated opinion data to participants, and allowing participants to reassess (and modify) their judgements after receiving feedback on the group opinions in order to guide a consensus-building process on the content validity of the measure.[38 39] The methods enable assessment of consensus between the panels, and points of divergence according to roles. During the surveys, experts will be asked to rate each item on the ORC according to its (1) relevance to chronic disease management in South Africa, (2) relevance to adoption of evidence-based practice, (3) relevance given health system resource constraints and (4) evaluability in South African PHC context. Two rounds of anonymous rating of the items and feedback on responses will be conducted. Any items achieving less than 70% consensus will be excluded.[40] This stage will be followed by a face-to-face expert meeting incorporating nominal group methods to discuss findings from phases I and II of the study and how these can be combined to shorten the TCU-ORC assessment. The combination of these methods has previously been shown to be useful for adapting a multidomain, multi-item, US-developed primary care assessment tool for the South African context.[40]

## Analyses

Analyses will follow the processes of similar validation studies.[15 41 42] STATA statistical software will be used for analysis and modelling. Multiple logistic regression analysis will be used to examine the association of demographic characteristics with TCU-ORC scores. Internal consistency of the TCU-ORC and subscales will be assessed by calculating Cronbach's alpha coefficients. For the retest sample, we will calculate the intraclass correlation coefficient to determine the test-retest reliability of the various scales. To assess the convergent validity of the ORC, correlation coefficients will be used to give evidence for correlation between the ORC, ORCA and CARI measures. To evaluate the construct validity of the TCU-ORC, we will examine the factor structure with both exploratory and confirmatory factor analysis. First, principal component analysis[43] with oblimin rotation will be used to examine the generalisablity of the factor structure of the original questionnaire to the South African data. Principal axis factoring will then be conducted to give further evidence of the factor structure.[43] Confirmatory factor analysis will be used to confirm whether the latent structure of the TCU-ORC identified in HIC is a good fit for the South African data.[43] Structural equation modelling techniques will be employed and acceptability of model fit data reported.

Based on the findings from these analyses, and to fulfil the secondary aim of the study, a shortened version of the TCU-ORC scale will be developed. Loading of items onto factors will be assessed to guide decisions around the relevance of each item and whether to retain each in the shortened tool. Data analysis from the administration of the instruments and from the Delphi process will be integrated into an approach for shortening the existing TCU-ORC scale.

## DISCUSSION

For some years, there has been discussion of how to reduce the MNS treatment gap in LMICs. Much of this gap in middle-income countries such as South Africa may be due to a lack of mechanisms for practically rolling out what is set out in national mental health policy.[44] This 'implementation gap' challenge is not limited to the field of mental health and hampers progress in HIC and LMIC in a variety of areas of public health. Identifying and addressing barriers to implementation is required to reduce the implementation gap and it is in this specific niche that this study aims to contribute. Our future aim is to use the tool developed in this study to identify and address barriers to implementation for the health intervention currently being assessed in the Project MIND trial.

The assessment of ORC as an avenue for addressing organisational roadblocks to implementation in health organisations is a relatively new but developing field of research in HIC. In LMIC, this area of research is in its infancy. Key learnings from the ORC literature from HIC

can be applied to hasten progress. This literature has identified the need to better assess ORC to identify and address barriers to implementation. The TCU-ORC has been identified as a potentially useful tool, and on face value it seems that the domains it measures are relevant for the South African healthcare context. For example, the motivation for change domain may be particularly relevant for use in this setting since PHC facility staff are constantly dealing with numerous competing priorities and challenges. As a result, the resolve to implement changes in relation to mental health services may be low. This would constitute an important barrier to adoption of a mental health intervention that, if identified, could be addressed before dedicating significant resources to implementation efforts. Resource and capacity constraints also may be significant obstacles to implementation of new services.[45] The TCU-ORC institutional resources and staff attributes domains may therefore yield key information on specific barriers or opportunities for implementation of such a service. Effective implementation efforts will need to harness available resources and skills and having a detailed picture of the resource and human capacity environment may be an important first step in this endeavour. It has also been suggested that organisations with more internally consistent staff ratings on the organisational climate domain have greater agreement among staff as to the functioning of the organisation and could be expected to be more likely to take on a change.[14] Items from this domain associated with support for the introduction of innovative practices in HIC contexts have included staff perceptions of greater need for improvements, greater impact of the influence of staff on each other and a stronger sense of 'organisational mission'.[45] The validation and contextual adaptation of this domain therefore may be important for understanding how primary healthcare organisations function, and whether their organisational climate enables change. This assessment would be of relevance for integration of MNS services into chronic care, as well as for adoption of any new complex intervention.

Although there has been little effort to investigate ORC in relation to chronic disease care and the integration of MNS services into chronic disease care in LMICs, there is evidently no 'one size fits all' solution for improving ORC in PHC facilities. The contextually validated ORC scale generated from this study will enable assessment of health facilities' strengths and weaknesses in relation to adoption of any new health practices, beyond the specific focus of MNS services. Lower mean scores on the ORC domains will indicate potential areas of weakness; higher mean scores will indicate potential strengths.[46] Understanding areas of good and poor performance will enable tailoring of implementation plans to address these strengths and weaknesses and consequently improve adoption and sustained implementation. Strategies to address barriers (eg, organisational culture barriers) may include providing information and effective participatory training on the new intervention, inclusive decision making with participation and discussion of all involved and addressing any concerns raised by front-line workers.[47] An important critique of adoption of heath innovations in general, particularly in the Sub-Saharan African context, is that plans for sustainability of innovations are not put in place from the outset.[48] Even in contexts where a new practice is readily adopted, ongoing effort is required to maintain delivery of innovative practices.[44] Failing to understand PHC organisations' 'functional dynamics'[14] and to support the process of organisational change through addressing barriers to change can reduce the likelihood of the change being sustained. A contextually appropriate ORCA tool may assist in identification of factors required to promote sustainability.

The development of an abbreviated tool for assessing ORC in this context is a key contribution to the field. Tools that have many items and are time consuming to complete are not as feasible or acceptable for use in under-resourced healthcare settings and are likely to be underutilised by busy health managers and staff. While balancing breadth and quality of information collected against brevity will be critical, an abbreviated tool could be a useful addition for facility, operational and district-level managers and others to 'diagnose' barriers to organisational change for adoption of new health interventions, not limited to mental health interventions. Research and development in this field requires input from health service staff, as much as academic researchers, in which both groups can contribute to understanding how to promote effective implementation and sustainability.[45] The planned Delphi process in this study may be a first step for developing this partnership.

This study has the potential to contribute to reducing the gap between availability of evidence-based practice and its implementation in routine care in South Africa, with potential generalisability to other LMICs. In practical terms, identification of barriers to change may enable strategic utilisation of limited resources to improve service delivery. The field of health system organisational change in LMIC has been hampered by a lack of validated measures that can be used to diagnose barriers and evaluate the impact of interventions to enhance ORC. By validating and abbreviating an ORC measure for an LMIC context, this study therefore has the potential to make an important contribution to the field of organisational change in the South African health system. While this work is being nested within Project MIND and the planned implementation of MNS services in primary care, it has potential for application in a wide variety of innovations, since the potential organisational barriers and facilitators to change are not specific to the Project MIND intervention or to MNS services.

Despite the relevance of this work to the PHC system in South African, there are several challenges to the collection of organisational data in this context. Possibly most important may be enabling health staff to find the time to complete the questionnaires and to ensure an adequate sample size. Second, the surveys will be presented in

English. Some health staff may have difficulty understanding some of the questions if English is not their first language. However, the official business language for health service employees in the Western Cape of South Africa is English, and textual adaptations can be made to make survey items more readily understandable to health workers whose home language is not English. A further methodological issue is that the ORC measure data will be collected only from participants in the better resourced Western Cape province which may limit its generalisability to other less-resourced provinces. Delphi study data will however be collected from participants in a number of provinces across the country and may enhance relevance and utility of findings to health providers from other parts of the country.

It is also important to consider that the TCU-ORC is a single instrument, and as such does not assess the broad scope of factors that impact on implementation. Well-established implementation frameworks such as the Consolidated Framework for Implementation Research[49] underscore the importance of other factors, such as the characteristics of the intervention itself, the characteristics of those who will receive and deliver the service and the implementation process. Efforts to enhance the implementation of MNS services in chronic disease care should bear these limitations in mind when examining barriers to implementation. In addition, this instrument measures readiness to adopt a service change in general, rather than readiness to adopt a specific change (such as the adoption of a new MNS service in chronic care). Existing evidence suggests that specific barriers to integration of mental health services in primary care (which would not be assessed by the TCU-ORC) may include contextual factors such as lack of mental health literacy, low prioritisation of mental illness, stigma towards service users with mental illnesses, low managerial and planning capacity, poor training and high levels of staff turnover among clinic staff.[6 12 50 51] In addition to assessing ORC, an understanding of these specific barriers will be key to supporting future integration of the task-shared MNS service. Additional tools and/or methods of assessment will be required to develop a full understanding of the likelihood of service implementation. Despite this caveat, assessing and addressing barriers to ORC as assessed by the TCU-ORC is an important first step to support the integration of MNS services into chronic care.

### Ethics and dissemination

In this study, there should be minimal risks to participants. No data or biological samples will be collected from patients. There is a small risk that questions on the ORC scale could raise discomfort for participants should they touch on issues causing stress or conflict in the workplace. Participation in the study will be voluntary and participants will be able to decline to participate. Participants will be provided with accessible information on the study and their participation. Informed consent will be sought from all participants in all aspects of the study. As participants will be health facility or district-level health staff, or academics and health system employees, the levels of literacy and education will be expected to be high enough to allow adequate understanding and completion of written consent forms. Participants will however be given a chance to discuss their participation prior to completing research instruments, or participating in the Delphi process. Anyone who does not wish to provide informed consent will be able to leave the study activity without negative consequences. Ethical approval for this study has been granted.

The close partnership with the Western Cape Department of Health and the focus on a health systems approach within this study give potential for dissemination throughout South Africa and other LMIC. The short-form ORCA tool as a tangible output of the study for use by health service managers will be a key tool to achieve this dissemination. Results of the study will be submitted to journals covering health systems strengthening and organisational change and implementation of evidence-based practices. Results will also be presented at national and international health systems and public health meetings.

### Future work

Several areas for future research are indicated. First, we plan to use the data from the ORCA process to design interventions to address barriers to adopting task-shared mental health programmes in South African facilities. This intervention strategy has been used for building readiness for providing care for depression in PHC in HIC contexts.[52] Second, we plan to use the ORC short form, developed through the proposed study, to evaluate the effect of the intervention to improve ORC to provide mental health interventions.

### CONCLUSION

While more flexible organisations (eg, small/medium-sized, non-hierarchical organisations) may find it easier to adopt changes, organisations that are more stable and controlled (such as health service organisations) may be better at ensuring sustainability of the services they offer.[13] This presents both a challenge and an opportunity in that introduction of service innovations in these organisations may be challenging but they may have a great return on investment if sustained in the long term. This lends importance to the assessment of ORC in health service organisations using contextually valid measures, such as those proposed in this study. Building readiness for change by addressing barriers as identified by this assessment promises to be a challenging but worthwhile endeavour for the future.

**Contributors** BM, CB-S and KS conceptualised the study. CL provided statistical support. PP-W advised on research processes for integration into the Project MIND trial. CB-S prepared the manuscript first draft. All authors edited and agreed on the final manuscript.

**Funding** This work was supported by the joint-funded initiatives of the British Medical Research Council, Wellcome Trust and DFID (MR/M014290/1) as well as funding from the South African Medical Research Council. CB-S is supported by the South African Medical Research Council Intramural Career Development Fund.

**Competing interests** None declared.

**Patient consent** Not required.

**Ethics approval** South African Medical Research Council Ethics Committee (Protocol ID EC004-2-2015,amendment of 20 August 2017) .

**Provenance and peer review** Not commissioned; externally peer reviewed.

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
