## [Reviewer comments · BMJ Open]

ARTICLE DETAILS

TITLE (PROVISIONAL)	Protocol for development and validation of a context-appropriate tool for assessing organisational readiness for change in primary health clinics in South Africa
AUTHORS	Brooke-Sumner, Carrie; Sorsdahl, Katherine; Lombard, Carl; Petersen-Williams, Petal; Myers, Bronwyn

VERSION 1 – REVIEW

REVIEWER	Bolanle Adeyemi Ola Lagos State University, Nigeria
REVIEW RETURNED	09-Dec-2017

GENERAL COMMENTS	It is a well written work.
----------------------------

REVIEWER	Abraham Wandersman University of South Carolina: United States
REVIEW RETURNED	22-Dec-2017

GENERAL COMMENTS	“When intra- organisational consensus on these factors is high, ORC can be said to be high, and individuals in the organisation are more likely to persist at and work together to effectively implement a change.” Consensus on the factors is not sufficient. There could be consensus that a factor related to readiness is low, and therefore making implementation less likely. You may also want to talk about what happens when there is no consensus, or when leadership had a different perspective than front line staff implementing the intervention. You plan on data about different positions; it may be necessary to do analysis by role. “Although there are several measures for assessing ORC, 121 a recent systematic 122 review highlighted that only seven of the 43 measures reviewed had evidence for their validity and reliability.” There is some more recent work to be aware of; notably “Evans, J. M., Grudniewicz, A., Baker, G. R., & Wodchis, W. P. (2016). Organizational Capabilities for Integrating Care: A Review of Measurement Tools. Evaluation & the health professions, 39(4), 391-420.” This is particularly relevant since counseling for CMD’s is being integrated into chronic disease care. “Further, the TCU-ORC in its current form is lengthy and time consuming to complete as it 139 comprises 125 items. The length of this tool reduces its utility for health managers and 140 providers in
---

	low resourced, high patient burden settings who have limited time to complete 141 organisational assessments (e.g. because of being drawn into clinical work due to high patient burden).” There is meta-issue in here worth discussing. If organizational managers can't devote 20-30 minutes to taking an assessment to help implement an initiative as comprehensive and complex as treating CMD in low resourced setting, isn't that a proxy indicator for not being ready at all? I think the goal should be not to reduce survey burden but to identify the highest quality information to inform program improvement. That may come from reduce items and other adaptation, but, in my opinion, should not be the primary purpose of reducing items. Line 241: Can you clarify what “feasibility of assessment” means? Is that evaluability?
--	--

REVIEWER	Allison J.ober RAND Corporation, USA
REVIEW RETURNED	27-Dec-2017

GENERAL COMMENTS	General Comments: The submitted article is a protocol paper describing a study that will develop and validate a short form of the TCU ORC for use in primary care clinics in LMIC. The paper is well-written, however, the authors may overstate the way in which the new tool can be used. Specifically, the ORC will not assess barriers specific to implementing MNS care in primary care settings in LMIC, rather it will may indicate weaknesses in areas thought to be associated with implementation of new practices in general. This is an important distinction that should be made throughout the manuscript. Nevertheless, as the authors point out, there is a great need in the field of implementation science for validated ORC tools that are applicable to a variety of contexts, thus the study will be an important contribution. Additional, more specific comments follow. Specific Comments: p. 2 line 54 Limitations: “Collection of survey data from only the Western Cape province may limit 54 generalizability, however the planned Delphi process will involve experts from 55 various provinces in the country.” Please describe differences between the Western Cape and other parts of the country that could limit generalizability within SA and in other LMIC. p. 3 lines 65, 72-77 Definition and use of MNS is confusing. On line 65, the authors define mental, neurological and substance use disorders as MNS. However, on line 72, they state that one quarter of people with a “mental disorder” gets treated, and on line 74, they include “substance use disorder” as a “common mental disorder.” The authors should define MNS, mental disorder, and substance use disorder consistently throughout the manuscript. As it currently reads, it is unclear whether one quarter of people with a mental disorder only, or a mental disorder with substance use, or an MNS, gets treated, and if substance use disorders are (and should be) thought of as CMDs. (This may be clearer to readers in South Africa, but international readers may need clearer definitions.)
--

p. 3 lines 80-88

Similar to the above comments, it is unclear here whether substance use disorder screening is included with mental health disorder screening, and if “brief mental health counseling” includes counseling for a substance use disorder. Please be clear at the start of the manuscript whether mental health services include substance use disorder services.

p. 3 lines 86-88

This description/definition of integrated mental health care could be expanded. Integrated care could span a range of possible levels and types of integration, such as co-located, partially integrated, fully integrated. See, for example, the article by Collins et al., “Evolving Models of Behavioral Health Integration in Primary care.”

P. 4 lines 103-105

The authors have drawn from Holt et al and Weiner et al for their definition of ORC. Although I agree that this definition (from Weiner and Holt) could possibly capture the sum of the elements thought to be critical to ORC, I am not certain if the definition the authors put forth quite captures what Holt and Weiner have described or how it relates to the other elements of readiness. Although definitions of organizational readiness vary widely, they have in common several key constructs, such as whether an organization’s culture and climate are ready to make general changes (for example, organizations with stronger staff morale, less staff turnover, and openness to new practices in general typically are more likely to support implementation of new practices), whether individual members view their organization as capable of change, or whether individual members are themselves prepared and willing to make a specific change or adopt a specific new practice. Holt and Weiner specifically refer to the last part—whether members are prepared and willing. This is similar to how the authors define ORC but a link needs to be made to the other elements of ORC. Specifically, is the assumption that if other elements of ORC are met (climate, culture, etc., as measured by the ORC), psychological and behavioral readiness (as defined by Holt and Weiner) will be higher? Although I generally support the authors’ definition of ORC, the TCU ORC scale does not specifically measure individual psychological or behavioral willingness/readiness to make a specific change the way it is described by Holt and Weiner, but instead measures generalized readiness for change. The authors should consider these issues in their definition. Finally, the ORC does not capture other elements of implementation thought to be important to adoption and sustainment. For example, the CFIR (Damschroder et al) includes a gamut of factors through to drive implementation that go well-beyond what the ORC measures. Although it is not expected that the authors should measure all of these factors, it is important to discuss the possible limitations of the constructs in the ORC to predicting sustained implementation.

p. 4 line 119

The word “in” is duplicated

p. 10 line 286-288

Many would argue that ORC is still in its infancy in HIC as well as LIC. Although I strongly agree with the authors that there is a need to measure ORC, there is also a need to come to consensus on the definition of ORC and ensure that the measures have content validity. One thing I am struck by is that the authors cite several

	barriers to integrated MNS care, including stigma and poor training on MNS care. These factors fit in the definition of those that might contribute to ORC, but are not measured by the TCU ORC. This is not to say that constructs within the TCU ORC are not important to implementation, but rather to again point out that a measure of general readiness to change does not address readiness to make a specific change, perhaps a limitation of the TCU ORC that the authors should point out. p. 10 lines 293-296 On a similar point, the ORC will not measure “urgency” to implement MNS care specifically, unless the authors plan to adapt the measure so it asks specifically about motivation to adoption MNS care. If this is a case, it should be noted in the methods section. Otherwise, what the ORC will measure is general motivation to change. If the authors wish to measure specific “barriers” to MNS integration, some other measure will be required. Again, this is not to say that the validation of the ORC to this context is not needed, but the authors should be clear on what the findings will mean and how they can be used. p. 10 lines 330-335 I agree that the strategies proposed by the authors might be effective for increasing implementation and sustainability of MNS care, but I am not convinced that the ORC will identify these as weaknesses. Again, and as noted by the authors, the ORC will measure general readiness to adopt a new practice but will not assess barriers that are specific to adoption of MNS care. The authors should be clear about how the ORC will help identify barriers, and perhaps should consider additional methods (beyond validation and use of the ORC) to measure barriers specific to MNS interventions. For this manuscript, the focus should be on how the ORC specifically will be used toward implementation of MNS interventions. p. 12 lines 370-372 As stated earlier, please elaborate on some of the key differences between the Western Cape and other provinces so international readiness can better understand context.
--	---

VERSION 1 – AUTHOR RESPONSE

Dear Dr Sucksmith,

Please find with this letter our revised manuscript entitled:

“Protocol for development and validation of a context-appropriate tool for assessing organisational readiness for change in primary health clinics in South Africa’

Thank you also to the reviewers for their thoughtful comments, which have improved the quality of the paper. We have amended the submission according to these comments. Please find below a detailed response and description of the changes made.

Sincerely
Carrie Brooke-Sumner

Reviewer: 2

“When intra- organisational consensus on these factors is high, ORC can be said to be high, and individuals in the organisation are more likely to persist at and work together to effectively implement a change.”

Consensus on the factors is not sufficient. There could be consensus that a factor related to readiness is low, and therefore making implementation less likely. You may also want to talk about what happens when there is no consensus, or when leadership had a different perspective than front line staff implementing the intervention. You plan on data about different positions; it may be necessary to do analysis by role.

Response: Thank you for this point. We agree that this sentence was confusing. We have reworded the paragraph from line 102 onwards to more clearly state the meaning of ORC as it relates to the context of the proposed study.

“Although there are several measures for assessing ORC, 121 a recent systematic 122 review highlighted that only seven of the 43 measures reviewed had evidence for their validity and reliability.”

There is some more recent work to be aware of; notably “Evans, J. M., Grudniewicz, A., Baker, G. R., & Wodchis, W. P. (2016). Organizational Capabilities for Integrating Care: A Review of Measurement Tools. *Evaluation & the health professions*, 39(4), 391-420.” This is particularly relevant since counselling for CMD’s is being integrated into chronic disease care.

Response: We have reviewed and now cited this reference. We have revised the text accordingly from line 133 to reflect the variety of other measures available.

“Further, the TCU-ORC in its current form is lengthy and time consuming to complete as it 139 comprises 125 items. The length of this tool reduces its utility for health managers and 140 providers in low resourced, high patient burden settings who have limited time to complete 141 organisational assessments (e.g. because of being drawn into clinical work due to high patient burden).”

There is meta-issue in here worth discussing. If organizational managers can’t devote 20-30 minutes to taking an assessment to help implement an initiative as comprehensive and complex as treating CMD in low resourced setting, isn’t that a proxy indicator for not being ready at all? I think the goal should be not to reduce survey burden but to identify the highest quality information to inform program improvement. That may come from reduce items and other adaption, but, in my opinion, should not be the primary purpose of reducing items.

Response: This is an important point. We agree that we should not be reducing the survey items purely for the sake of time taken. However in the South African context we do feel that a shortened survey is more feasible given that facility managers (as clinicians) do get drawn into clinical work, leaving less time for administrative tasks. We have reworked the section from line 148 to reflect this.

Line 241: Can you clarify what “feasibility of assessment” means? Is that evaluability?

Response: In this context we mean the degree to which the item can be assessed using a survey scale methodology by primary health facility staff/managers, so yes evaluability. We have revised accordingly.

Reviewer: 3

General Comments:

The submitted article is a protocol paper describing a study that will develop and validate a short form of the TCU ORC for use in primary care clinics in LMIC. The paper is well-written, however, the authors may overstate the way in which the new tool can be used. Specifically, the ORC will not assess barriers specific to implementing MNS care in primary care settings in LMIC, rather it will indicate weaknesses in areas thought to be associated with implementation of new practices in general. This is an important distinction that should be made throughout the manuscript. Nevertheless, as the authors point out, there is a great need in the field of implementation science for validated ORC tools that are applicable to a variety of contexts, thus the study will be an important contribution.

Response: This point regarding the scope of the use of the tool is well taken – we have revised throughout the manuscript accordingly, to indicate that the tool may be used to identify barriers to implementation of new practices in general. But we have retained the information regarding the context in which the tool is being developed (i.e. mental health services), and our plans to use the tool to support implementation of this task shared mental health service.

Specific Comments:

p. 2 line 54

Limitations:

“Collection of survey data from only the Western Cape province may limit generalizability, however the planned Delphi process will involve experts from 55 various provinces in the country.” Please describe differences between the Western Cape and other parts of the country that could limit generalizability within SA and in other LMIC.

Response: The main difference between the Western Cape and other parts of the country is that the Western Cape is generally better resourced in terms of mental health care services. This has been clarified.

p. 3 lines 65, 72-77

Definition and use of MNS is confusing. On line 65, the authors define mental, neurological and substance use disorders as MNS. However, on line 72, they state that one quarter of people with a “mental disorder” gets treated, and on line 74, they include “substance use disorder” as a “common mental disorder.” The authors should define MNS, mental disorder, and substance use disorder consistently throughout the manuscript. As it currently reads, it is unclear whether one quarter of people with a mental disorder only, or a mental disorder with substance use, or an MNS, gets treated, and if substance use disorders are (and should be) thought of as CMDs. (This may be clearer to readers in South Africa, but international readers may need clearer definitions.)

Response: We have revised the section beginning line 73 so that we refer only to MNS disorders, which clarifies the issues raised.

p. 3 lines 80-88

Similar to the above comments, it is unclear here whether substance use disorder screening is included with mental health disorder screening, and if “brief mental health counselling” includes counselling for a substance use disorder. Please be clear at the start of the manuscript whether mental health services include substance use disorder services.

Response: We have revised from line 80 onwards to reflect that the screening and counselling referred to in the study is for disorders such as depression and alcohol use disorder

p. 3 lines 86-88

This description/definition of integrated mental health care could be expanded. Integrated care could span a range of possible levels and types of integration, such as co-located, partially integrated, fully integrated. See, for example, the article by Collins et al., "Evolving Models of Behavioral Health Integration in Primary care."

Response: We have revised this sentence to explain what the integration of this service means in the South African context.

P. 4 lines 103-105

The authors have drawn from Holt et al and Weiner et al for their definition of ORC. Although I agree that this definition (from Weiner and Holt) could possibly capture the sum of the elements thought to be critical to ORC, I am not certain if the definition the authors put forth quite captures what Holt and Weiner have described or how it relates to the other elements of readiness. Although definitions of organizational readiness vary widely, they have in common several key constructs, such as whether an organization's culture and climate are ready to make general changes (for example, organizations with stronger staff morale, less staff turnover, and openness to new practices in general typically are more likely to support implementation of new practices), whether individual members view their organization as capable of change, or whether individual members are themselves prepared and willing to make a specific change or adopt a specific new practice. Holt and Weiner specifically refer to the last part—whether members are prepared and willing. This is similar to how the authors define ORC but a link needs to be made to the other elements of ORC. Specifically, is the assumption that if other elements of ORC are met (climate, culture, etc., as measured by the ORC), psychological and behavioural readiness (as defined by Holt and Weiner) will be higher? Although I generally support the authors' definition of ORC, the TCU ORC scale does not specifically measure individual psychological or behavioural willingness/readiness to make a specific change the way it is described by Holt and Weiner, but instead measures generalize readiness for change. The authors should consider these issues this in their definition. Finally, the ORC does not capture other elements of implementation though to be important to adoption and sustainment. For example, the CFIR (Damschroder et al) includes a gamut of factors through to drive implementation that go well-beyond what the ORC measures. Although it is not expected that the authors should measure all of these factors, it is important to discuss the possible limitations of the constructs in the ORC to predicting sustained implementation.

Response: We have revised the paragraph from line 101 to cover the suggestions of the reviewer, going more in depth into the definitions of Weiner and Holt, and also linking the individual aspects of ORC to the organisational aspects

Response: We have added a paragraph in the discussion, from line 401, to discuss other factors with a bearing on implementation and the need for their assessment.

p. 10 line 286-288

Many would argue that ORC is still in its infancy in HIC as well as LIC. Although I strongly agree with the authors that there is a need to measure ORC, there is also a needed to come to consensus on the definition of ORC and ensure that the measures have content validity. One think I am struck by is that the authors cite several barriers to integrated MNS care, including stigma and poor training on MNS care. These factors fit in the definition of those that might contribute to ORC, but are not measured by the TCU ORC. This is not to say that constructs within the TCU ORC are not importation to implementation, but rather to again point out that a measure of general readiness to change does not

address readiness to make a specific change, perhaps a limitation of the TCU ORC that the authors should point out.

Response: We agree with this comment and have now addressed it in the paragraph above added to the discussion.

p. 10 lines 293-296

On a similar point, the ORC will not measure “urgency” to implement MNS care specifically, unless the authors plan to adapt the measure so it asks specifically about motivation to adoption MNS care. If this is a case, it should be noted in the methods section. Otherwise, what the ORC will measure is general motivation to change. If the authors wish to measure specific “barriers” to MNS integration, some other measure will be required. Again, this is not to say that the validation of the ORC to this context is not needed, but the authors should be clear on what the findings will mean and how they can be used. This point has been addressed by the above paragraph in the discussion.

p. 10 lines 330-335

I agree that the strategies proposed by the authors might be effective for increasing implementation and sustainability of MNS care, but I am not convinced that the ORC will identify these as weaknesses. Again, and as noted by the authors, the ORC will measure general readiness to adopt a new practice but will not assess barriers that are specific to adoption of MNS care. The authors should be clear about how the ORC will help identify barriers, and perhaps should consider additional methods (beyond validation and use of the ORC) to measure barriers specific to MNS interventions. For this manuscript, the focus should be on how the ORC specifically will be used toward implementation of MNS interventions.

Response: We have included in the paragraph in the discussion the need for other methods to assess specific barriers to MNS integration.

VERSION 2 – REVIEW

REVIEWER	Abraham Wandersman University of South Carolina, USA
REVIEW RETURNED	19-Feb-2018

GENERAL COMMENTS	While you cite the Evans et al paper about survey length, and see that you all noted the tradeoffs between quality and brevity, I'm not convinced that 60 items is a meaningful threshold when the idea is to put the best quality implementation forward. A debate about this seems to be outside of the scope of this paper, so I am satisfied with how you addressed my earlier comments.
--

REVIEWER	Allison J. Ober RAND Corporation, USA
REVIEW RETURNED	09-Mar-2018

GENERAL COMMENTS	The submitted article is a revision of a protocol paper describing a study that will develop and validate a short form of the TCU ORC for use in primary care clinics in LMIC. The authors have addressed one of the main concerns with the original submission – that the ORC will not assess barriers specific to implementing MNS care in primary care settings in LMIC, rather it will may indicate weaknesses in areas thought to be associated with implementation of new practices in general. The authors have also satisfactorily addressed other concerns raised by reviewers. The paper will set the stage for
---

	subsequent articles on the development and validation of the TCU ORC short form for use in these settings.
--	--